# Infection and Persistence of *Coxiella burnetii* Clinical Isolate in the Placental Environment

**DOI:** 10.3390/ijms24021209

**Published:** 2023-01-07

**Authors:** Sandra Madariaga Zarza, Muriel Militello, Laetitia Gay, Anthony Levasseur, Hubert Lepidi, Yassina Bechah, Soraya Mezouar, Jean-Louis Mege

**Affiliations:** 1MEPHI, IRD, APHM, Aix-Marseille University, 13005 Marseille, France; 2Institue Hospitalo, Universitaire Mediterranée Infection, 13005 Marseille, France; 3Immunology Department, Assitance Publique Hopitaux de Marseille (APHM), 13005 Marseille, France

**Keywords:** *Coxiella burnetii*, pregnancy, placenta, trophoblasts, placental macrophages

## Abstract

Infection by *Coxiella burnetii*, the etiological agent of Q fever, poses the risk of causing severe obstetrical complications in pregnant women. *C. burnetii* is known for its placental tropism based on animal models of infection. The Nine Mile strain has been mostly used to study *C. burnetii* pathogenicity but the contribution of human isolates to *C. burnetii* pathogenicity is poorly understood. In this study, we compared five *C. burnetii* isolates from human placentas with *C. burnetii* strains including Nine Mile (NM) as reference. Comparative genomic analysis revealed that the Cb122 isolate was distinct from other placental isolates and the *C. burnetii* NM strain with a set of unique genes involved in energy generation and a type 1 secretion system. The infection of Balb/C mice with the Cb122 isolate showed higher virulence than that of NM or other placental isolates. We evaluated the pathogenicity of the Cb122 isolate by in vitro and ex vivo experiments. As *C. burnetii* is known to infect and survive within macrophages, we isolated monocytes and placental macrophages from healthy donors and infected them with the Cb122 isolate and the reference strain. We showed that bacteria from the Cb122 isolate were less internalized by monocyte-derived macrophages (MDM) than NM bacteria but the reference strain and the Cb122 isolate were similarly internalized by placental macrophages. The Cb122 isolate and the reference strain survived similarly in the two macrophage types. While the Cb122 isolate and the NM strain stimulated a poorly inflammatory program in MDM, they elicited an inflammatory program in placenta macrophages. We also reported that the Cb122 isolate and NM strain were internalized by trophoblastic cell lines and primary trophoblasts without specific replicative profiles. Placental explants were then infected with the Cb122 isolate and the NM strain. The bacteria from the Cb122 isolate were enriched in the chorionic villous foetal side. It is likely that the Cb122 isolate exhibited increased virulence in the multicellular environment provided by explants. Taken together, these results showed that the placental isolate of *C. burnetii* exhibits a specific infectious profile but its pathogenic role is not as high as the host immune response in pregnant women.

## 1. Introduction

*Coxiella burnetii* is an intracellular bacterium and the causative agent of Q fever. Infected individuals are usually men of a mature age. They are mostly asymptomatic (60%), while 40% develop an acute Q fever and 2% progress to a persistent focal infection [1]. During pregnancy, exposure to *C. burnetii* increases the risk of developing acute Q fever but the illness is most often asymptomatic, which delays a diagnosis based on serology [2,3,4,5,6]. Hence, obstetrical complications including miscarriage, malformations, growth retardation, premature delivery and foetal death may occur [7,8,9]. The dominant point of view about the pathophysiology of Q fever is the importance of the host’s immune response to the detriment of *C. burnetii* strains [1,4]. Although the presence of the QpDV plasmid in *C. burnetii* strains is associated with an increased risk of abortion [10,11], the pathophysiology of *C. burnetii* infection during pregnancy remains unclear in humans, unlike in animals. *C. burnetii* infects domestic ruminants, leading to infertility, endometritis and mastitis, and obstetrical complications such as still birth and abortion (90%) [12,13]. The pathogenesis of *C. burnetii* obstetrical complications in animal models results from placental infection and the recruitment of immune cells [2,3,13,14]. The role of *C. burnetii* organisms is counteracted by the benefits of antibiotic treatments and the vaccination strategy that have been proposed to reduce abortion rates [14,15].

It has been shown that the reference Nine Mile (NM) strain of *C. burnetii* infects trophoblasts, the major cell population of placenta, with different degrees of differentiation. Indeed, NM infects the villous trophoblast BeWo cells and extra-villous trophoblast JEG-3 cells [16]. Progesterone, a major hormone of pregnancy produced by trophoblast cells, had been suggested as a relevant factor in *C. burnetii* pathogenesis and pathogen replication in placental cells [17]. Indeed, the addition of progesterone to extra-villous trophoblast JEG-3 cells inhibits *C. burnetii*-containing vacuole development and bacterial persistence. Additionally, oestrogens, which have a key role during pregnancy, play a protective role in macrophage response during *C. burnetii* infection [18,19]. The *C. burnetii* NM-infected BeWo cell line presents a specific transcriptomic program with modulated genes involved in apoptosis, cell motility, cell–cell signalling, and immune and inflammatory responses [16]. As macrophages are considered natural target cells for *C. burnetii* [20], we investigated bacterial infection and replication in monocyte-derived macrophages (MDM) and placental macrophages. *C. urnetiid* NM is also able to infect isolated placental macrophages (PM) from healthy at-term placentas [21]. After infection, the bacteria are progressively eliminated by PMs until nine days post-infection. Interestingly, this elimination is related to the release of interferon (IFN)-γ associated with a powerful and specific pro-inflammatory phenotype (M1 polarization profile) [22].

To date, it is not known whether *C. burnetii* placental cell infections alone account for host susceptibility, or whether bacterial isolates specifically contribute to placental pathogenesis. As almost all pathophysiological studies of Q fever during pregnancy used an NM reference strain that is far from real infection conditions, we isolated placental isolates of *C. burnetii* and compared them to the NM reference strain. The comparative genomic analysis and mouse infection revealed the specificity of the Cb122 placental isolate. We showed that the Cb122 isolate infected macrophages including PMs and trophoblasts in a similar way to the NM strain reference. In placental explants, the Cb122 isolate exhibited tropism for foetal tissues. Taken together, these results provided the evidence that *C. burnetii* organisms play a role in the pathophysiology of Q fever during pregnancy but the host immune response remains prominent.

## 2. Results

### 2.1. Comparative Genomic Analyses of Placental C. burnetii Isolates

Regarding genomic features, the whole-genome size of *C. burnetii* strains ranged from 1,995,457 to 2,093,477 bp (Table 1). The pangenome size delineated for these nine strains reached 3446 genes comprising clusters or unique genes. The genome-based phylogenetic tree showed that *C. burnetii* organisms were divided into three branches (Figure 1A). Placental isolates (Cb93, Cb51 and Cb87) were grouped in the phylogenetic tree whereas two other strains were distantly related (Cb122 and Cb48). Both strains Cb175 and RSA493 were close to the Cb93, Cb51 and Cb87 group as compared to the Cb122 and Cb48 group.

The pan-genome composition is shown in Figure 1B. Unique genes were predominantly retrieved in the placental isolates Cb48, Cb93 and Cb122, (38, 93 and 44 genes, respectively). When compared to reference strains, isolates Cb93 and Cb122 showed a greater number of unique genes in comparison to the other three placental isolates, 93 and 44, respectively (Appendix A).

A total of 3446 genes were identified and 1382 genes were present in the core genome, with 40% sequence identity among the species (Appendix A). The detailed list of genes in the pan-genome analysis is provided in the Appendix A. All placental isolates shared 44.2% of genes with 2.1% unique for placental isolates. They also lacked 25.3% of genes from all gene analyses with reference strains. The distantly related strain Cb122 contained 44 unique genes including 84% of hypothetical protein. Functional annotation was assigned for seven unique genes of the Cb122 isolate (16%). Interestingly, among these unique genes, we reported alpha-hemolysin translocation ATP-binding protein (HlyB), CRISPR-associated endonuclease Cas1, glutathione import ATP-binding protein (GsiA), xylulose kinase, sulfurtransferase and two IS family transposase.

We reported alpha-hemolysin translocation ATP-binding protein (HlyB) as a candidate protein. Homology searches were carried out using NCBI’s Conserved Domain Database (CDD) [23]. According to subfamily domain architectures, the candidate protein (Cb122_00594) belongs to the ABC_6TM exporters’ superfamily (cl38913, E-value 6.31 × 10^−27^). This family represents a subunit of transmembrane (TM) helices typically found in the ATP-binding cassette (ABC) transporters known as exporters (Appendix A). A detailed annotation of *C. burnetii* isolates is shown in Appendix A.

Moreover, protein modelling of CB122_00594 was performed [24]. In total, 174 residues have been modelled with 99.9% confidence by the single highest scoring template. Tridimensional prediction of the Cb122_00594 protein confirmed the structure of a hemolysin of a secretion system hlyb/d complex, with significant scores (confidence of 99.94 and E-value = 1.1 × 10^−24^) (Appendix A).

Taken together, the Cb122 isolate exhibited genomic originality compared to the NM strain and further experiments were performed with the two *C. burnetii* organisms.

### 2.2. C. burnetii Placental Isolate Virulence

Because BALB/c mice have been used to model acute human *C. burnetii* infection [25], we infected them with 1 × 10^8^ bacteria by the *intraperitoneal* route for 11 days. We found that their body weight decreased slightly following infection with the NM reference strain and decreased markedly after infection with the Cb122 isolate (day 1 *p = 0.0098*, day 4 *p* < 0.0001, day 8 *p* < 0.0001 and day 11 *p* < 0.0001) (Figure 2A).

The response of mice to infection was studied as follows. First, the spleen size was determined. The NM strain induced an increase in its size, which was largely enhanced in response to the Cb122 isolate (*p =* 0.0281) (Figure 2B). Second, the titer of antibodies directed against *C. burnetii* was determined. No differences between the titers of specific IgM and IgG were observed between the NM bacteria and the placental isolate (Appendix A).

Finally, tissue infection was measured at day 7 p.i. using qPCR [26]. Bacteria were detected in the blood, lungs, spleen, liver, adipose tissue and heart (Figure 2C). The number of *C. burnetii* DNA copies in the adipose tissue, blood and spleen was not significantly different in mice infected with NM and Cb122 placental isolates. The number of DNA copies in the lung, liver, adipose tissue and heart was higher in mice infected with the Cb122 isolate than in those infected with NM (*p =* 0.0079). In contrast, the infection profile of other placental isolates was similar to that of the NM strain (Appendix A). Taken together, these results suggested that the placental Cb122 isolate was more virulent than the reference strain *C. burnetii*.

### 2.3. Macrophage Response to Placental C. burnetii Isolate

As it is well established that *C. burnetii* infects and survives within monocytes and macrophages [27,28], we wondered if Cb122 placental isolates behave differently from the NM strain. We measured the infection of monocytes and MDMs by these two organisms. First, the uptake of *C. burnetii* organisms (50 bacteria per cell) by monocytes and MDMs was evaluated by immunofluorescence and qPCR (Figure 3A). Compared to the NM strain, the Cb122 isolate was similarly internalized by monocytes. In contrast, the Cb122 placental isolate was less internalized than the reference strain by MDMs (*p* < 0.0001). Nevertheless, the survival profiles of the NM strain and Cb122 placental isolate were similar in each cell. *C. burnetii* organisms were not eliminated by monocytes and exhibited a discrete replication in MDM (Figure 3B).

We then investigated the ability of *C. burnetii* organisms to infect PMs. We previously reported that the NM strain was internalized by macrophages from at-term placenta but were steadily eliminated [22]. We found here that the Cb122 placental isolate infected PMs (Figure 4A) as did the NM strain (Figure 4B). Similarly, the organisms moderately replicated in PMs within a 12-day period (Figure 4C) as observed above with MDM (Figure 3B).

All together, these results highlight that the Cb122 placental isolate and NM survive in monocytes and replicate in MDMs and PMs to a lesser extent. However, the Cb122 placental isolate did not exhibit a specific pattern of infection in macrophages.

As we previously reported that the infection of macrophages is closely related to their inflammatory state [20], we investigated the expression of three inflammatory cytokines, *TNF*, *IL1* and *IL6*, and one immunoregulatory cytokine, *IL10*, by MDMs and PMs infected with *C. burnetii* using qRT-PCR. MDMs and PMs infected with NM and Cb122 placental isolate stimulated the expression of *TNF*, *IL1B*, *IL6* and *IL10* (*p =* 0.0099) (Figure 5A). However, the expression of the four genes in PM-infected *C. burnetii* organisms was higher than in infected MDM cells. The evaluation of the cytokine release showed overall higher TNF, IL-6 ad IL-1β levels in PMs stimulated by *C. burnetii* NM and Cb122 isolates than in MDMs (Figure 5B). In contrast, IL-10 release was higher in MDMs infected with *C. burnetii* isolates than in PMs. Again, we did not observe a difference between the NM response and the Cb122 isolate response. Altogether, these results illustrated that the Cb122 isolate *C. burnetii* induces a greater inflammatory response in PMs compared to MDMs. This greater inflammatory profile gives PMs the ability to limit *C. burnetii* infection whereas the secretion of IL-10 observed by infected-MDMs may reflect their permissiveness.

### 2.4. Placental C. burnetii Isolate Specificity for Placental Tissue

To account for the tropism of the *C. burnetii* isolate for placenta, we investigated the ability of the Cb122 isolate *C. burnetii* to infect trophoblasts, the major placental cells. In an initial series of experiments, we infected villous BeWo trophoblast cells and extra-villous JEG-3 cells with the Cb122 placental isolate and NM strain. The uptake of *C. burnetii* organisms as determined by PCR was similar in villous trophoblast BeWo cells and in extra-villous JEG-3 cells (Figure 6A). While the number of bacterial copies remained similar in BeWo and JEG-3 cells infected with the NM strain, the two cell lines exhibited a discrete replication of the Cb122 placental isolate. Next, primary trophoblasts from full-term placentas were isolated and the bacterial uptake was measured. The number of DNA copies of Cb122 placental isolates was similar to that of NM as previously observed for placental cell lines (Figure 3A). Regarding the survival/replication of the placental isolate, we did not observe a replication profile (Figure 6B). Hence, trophoblastic cells were not more permissive for *C. burnetii* organisms than macrophages.

We hypothesized that placental tropism cannot be assessed in cells outside of their microenvironment. Hence, we infected placental explants from healthy controls with the Cb122 isolate of *C. burnetii* and NM strain and we measured the number of DNA copies in decidua basalis (DB), inner chorionic villous (iCV) and chorionic villous on the foetal (CVF) side. (Figure 7A). The number of DNA copies in DB and iCVM was similar in response to the Cb122 isolate and NM strain. In CVF, levels of Cb122 isolate DNA copies were higher than that of the NM strain (*p = 0.0005*) (Figure 7B). Interestingly, whereas the NM strain DNA copies slightly decreased during the 8-day culture, the Cb122 isolate copies remained constant during the same period of time in the three zones of the explant (Figure 7C). Hence, the Cb122 isolate of *C. burnetii* exhibited specificity for the foetal side of the placenta.

## 3. Discussion

Pregnancy increases the susceptibility of women to *C. burnetii*, which poses the risk of mothers developing obstetrical complications. Because of the anti-inflammatory context of pregnancy, the symptoms of Q fever are attenuated, thus delaying patient care [4]. In addition, the acquisition of Q fever increases the risk of maternal persistent infection manifesting as endocarditis [29]. The respective role of bacterium and/or host response in Q fever pathophysiology has been long debated [4]. The mechanisms of *C. burnetii* infection during pregnancy are poorly understood. It has been suggested that the Netherlands strain responsible for a large Q fever outbreak in the Netherlands may cause obstetrical complications and spontaneous abortion [5]. E. Angelakis et al. showed that the QpDV plasmid was more frequent in isolates associated with abortion, which suggests a strain-specific effect on Q fever prognosis in pregnancy via genotypic analysis [8]. This is the first time that the role of *C. burnetii* strains isolated from placenta from women with Q fever has been assessed in vivo and ex vivo.

Firstly, we analysed the sequencing data of five placental isolates of *C. burnetii* and we compared them with NM, reference strain, and three other strains (Netherlands, Guyana and Germany). The pan genome analysis showed that placental isolates share a large proportion of genes and the greater number of unique genes was found in Cb93 and Cb122 isolates of *C. burnetii*. Our study focused on the Cb122 strain, as this isolate appeared as a unique branch of the genomic-based phylogenetic tree of the nine investigated strains. Among the genomic specificities of the Cb122 isolate, we reported the presence of an alpha-hemolysin translocation ATP-binding protein (HlyB). This protein is part of the ABC transporter complex involved in energy generation and in the type 1 secretion system [30]. Previous papers reported specific genomic features in virulent strains associated with the deletion of the operon hlyCABD in the T1SS [31]. Furthermore, another energy-coupling protein, the glutathione import ATP-binding protein, was identified among the unique genes of Cb122. Transcriptomic analyses of these genes during the different steps of cell infection could be carried out to highlight their functional roles. Experimental investigation of these candidate proteins has to be performed to decipher their functional involvement in the virulence of this isolate. The specificity of this strain was emphasized by infection profile of Balb/C mice. As compared to NM and other placental strains, the Cb122 strain more significantly affected body weight and spleen size, two markers of infection, and was more effective in infecting tissue. Hence, the Cb122 strain exhibited a greater infection ability than the other strains of *C. burnetii*, thus suggesting a higher virulence of this placental isolate.

Secondly, we and other teams have shown that *C. burnetii* infects and survives within myeloid cells by interfering with the uptake mechanism and intracellular traffic [20,32,33]. In pregnancy, macrophages are enriched in placental lesions and can be infected. The intranasal inoculation of pregnant goats with the Netherlands isolate led to an inflammatory response in allantochorionic stroma consisting of neutrophils and macrophages [34]. We previously reported that *C. burnetii* NM infects isolated placental macrophages. Indeed, these PMs represent a mixed population of maternal and foetal origin. *C. burnetii* is eliminated by these macrophages as they induce an inflammatory program in which IFN-γ is critical [22]. We showed here that the placental isolate behaves differently from the NM strain in MDM but not in monocytes and PMs, suggesting that these strains and isolates use different receptors according to the cell type. Surprisingly, the placental isolate was less internalized by MDM than the NM strain. This phenomenon was observed when we compared *C. burnetii* NM phase I and II and we concluded that the low phagocytosis was a hallmark of virulence in macrophages [27]. The low phagocytosis of phase I *C. burnetii* was the consequence of uncoupling CR3 from αvβ3 integrin, which is known to recognize the bacterium. If the cooperation between CR3 and αvβ3 integrin is forced, the efficiency of phagocytosis is restored and virulent bacteria are eliminated [35]. We suggest that similar mechanisms account for the response to the placental isolate. However, these results do not support the hypothesis that placental macrophages are specifically targeted by placental isolates. They replicate in MDM and PMs more efficiently than in monocytes, which is in line with initial papers [36,37]. This was supported by the evidence of the poorly inflammatory phenotype of macrophages that enables bacterial replication. Here, we found that MDM overexpresses IL-10 which is required for *C. burnetii* replication, in contrast to PMs that exhibit a proinflammatory phenotype. This in line with S. Mezouar’s paper which reported the role of IFN-γ in the microbicidal competence of PMs towards *C. burnetii* [22]. The inability of *C. burnetii* to establish a replicative niche in PMs compared to other tissue macrophages explains why the complications are relatively rare in Q fever. Hence, resident alveolar macrophages are susceptible to *C. burnetii* infection, which accounts for the frequency of pneumonia in Q fever [38].

As trophoblasts have been reported to be infected with *C. burnetii* in placenta from patients and in infected animals, we wondered if the potential tropism of *C. burnetii* for placenta was due to a better interaction with trophoblasts. We found that placental isolates were less efficiently internalized than the reference strain by primary trophoblasts isolated from at-term placenta compared to those reported for MDM. However, they survived without real replication, suggesting that trophoblasts may be a niche for *C. burnetii* independent of the origin of the microorganisms. The use of placental cell lines provided results that can be carefully extrapolated to human pregnant women. Hence, we reported that *C. burnetii* NM replicated in human villous BeWo cells [16]. Placental isolates exhibit a proliferative profile in the BeWo cell line. Distinct results were obtained with the extravillous JEG-3 cell line which behaves as a primary trophoblast. These findings suggested that isolated trophoblasts cannot be a replicative milieu for *C. burnetii* even in placental isolates.

The observation of infected placentas in which trophoblasts are infected with *C. burnetii* seems to contradict data from isolated trophoblasts. The cooperation between decidual macrophages and trophoblasts is known to control the functions of the two cell types. Hence, M2 macrophages regulate the migration and invasion of trophoblasts [39]. Conversely, trophoblasts induce macrophage polarization to the M2 subtype [40]. The use of placental explants integrates the cooperation between immune and non-immune placental cells and the areas of maternal and foetal origin. The model of placental explants has been largely used in viral infections such as cytomegalovirus, Zika and more recently Severe Acute Respiratory Syndrome Coronavirus-2 [41,42,43]. The ability of *C. burnetii* to infect placental explants was not known. We found that *C. burnetii* NM infects at-term placental explants but was slightly eliminated. In contrast, the Cb122 placental isolate was more internalized than the reference strain, specifically in the chorionic villous foetal side. Regardless of location, the Cb122 placental isolate was not eliminated. Hence, placental explants were the only configuration in which the difference between placental isolate and reference strain was clearly observed.

In conclusion, the placental isolate exhibits a relatively specific infection profile which is not sufficient to explain the pathophysiology of Q fever during pregnancy. According to the data obtained with *C. burnetii* strains with varying degrees of virulence, it is their ability to interfere with the host immune response that is critical. The inflammatory response induced by *C. burnetii* in PMs accounts for their resistance to the microorganism, but it is not sufficient to interfere with the success of pregnancy. Further experiments with placental explants and mouse models of pregnancy will be necessary to understand the pathogenicity of *C. burnetii* during pregnancy.

## 4. Materials and Methods

### 4.1. Ethics Statement

Nineteen full-term human placentas were collected at the Gynaecology-Obstetric Department of the “Hôpitaux de la Conception” hospital (Marseille, France) with the informed consent of the mothers and following the study being approved by the ethics committee of Aix-Marseille University (convention No.08–012).

Blood samples (Leucopacks) were obtained from the “Etablissement français du sang” blood service, which carried out donor inclusions, informed consent, and sample collection (convention n°7828).

The animal experimental protocol (APAFIS #18280) was approved by the ethics committee “C2EA-14” of Aix-Marseille University (Marseille, France) and the French Ministry of National Education, Higher Education and Research.

### 4.2. Placenta Collection and Primary Cell Cultures

Placentas were collected after delivery from healthy pregnant woman with a gestational age of 36–42 weeks. After removing the umbilical cord, membrane tissues were digested using Hank’s Balanced Salt Solution (HBSS, Life Technologies, Carlsbad, CA, USA), 2.5 mM DNase I (Sigma-Aldrich, Saint-Quentin-Fallavier, France) and 2.5% trypsin (Life Technologies).

Primary trophoblasts were isolated from full-term placenta after performing a Percoll (VWR) cushion and an Epithelial Growth Factor Receptor (EGFR)^+^ selection using specific antibodies (Anti-EGFR-IgG2a, TEBU BIO, Le Perray-en-Yveline, France) and magnetic beads targeting IgG2a antibodies (Miltenyi Biotec, Bergisch Glabach, Germany). The purified EGFR+ trophoblasts were cultured for 24 h in DMEM F-12 Ham medium containing 10% foetal bovine serum (FBS), 2 mM L-glutamine, 100 U/mL penicillin and 50 μg/mL streptomycin (Life Technologies) at 37 °C, 5% CO2. More than 95% of the cells were EGFR^+^, as assessed by flow cytometry.

PMs were isolated after performing a Ficoll cushion and CD14^+^ selection, as previously described [22]. The purified CD14^+^ PMs were cultured in DMEM F-12 Ham medium (Life Technologies) containing 10% FBS, 2 mM L-glutamine and antibiotics for 16 h. More than 98% of purity was observed, as assessed by flow cytometry.

Human BeWo (villous trophoblast, clone CCL-98) and JEG-3 (extra-villous trophoblast, clone HTB-36) were obtained from the American Type Culture Collection (ATCC, Manassas, VA, USA). Trophoblast cell lines were cultured in DMEM F-12 Ham medium supplemented with 10% FBS, 2 mM L-glutamine and antibiotics, and cultured at 37 °C, 5% CO_2_.

Peripheral blood mononuclear cells (PBMCs) were isolated from buffy coats by centrifugation on a Ficoll cushion (Eurobio, Les Ulis, France). Monocyte cells were obtained from PBMCs after incubating cells in Roswell Park Memorial Institute-1640 medium (RPMI, Life Technologies) containing 20 mM HEPES (Life Technologies), 2 mM L-glutamine, 100 U/mL penicillin and 50 μg/mL streptomycin for 2 h at 37 °C, 5% CO_2_. After washing with PBS, non-adherent cells were removed and adherent cells were designated as monocytes as 95% of the cells expressed CD14, a monocyte marker, as assessed by flow cytometry. Monocyte cells were cultured in RPMI containing 10% FBS, 20 mM HEPES, 2 mM L-glutamine, 100 U/mL penicillin and 50 µg/mL streptomycin for 16 h before infection. Monocytes were then differentiated into macrophages (MDMs) by incubating them in RPMI 1640 containing 10% of heat-inactivated human AB serum (MP Biomedicals, Solon, OH, USA), 2 mM glutamine and antibiotics for 3 days and then in RPMI 1640 supplemented with 10% FBS for 4 additional days, as previously described [18]. More than 95% of the differentiated cells were MDMs, as determined by the expression of CD68.

### 4.3. C. burnetii Isolation and Cell Infection

Five *C. burnetii* placental isolates, including Cb48, Cb51, Cb87, Cb93 and Cb122, were evaluated in comparison with the reference Nine Mile (NM) RSA493 strain in phase I (Table 2). *C. burnetii* isolates and the NM strain were cultured in MRC5 for no more than three passages to keep the bacteria in phase I. Cells were sonicated, then centrifuged, and bacterial pellets were recovered and kept at −80 °C as previously described [16]. The concentration of bacteria was determined by Gimenez staining and bacterial viability using the Live/Dead BacLight bacterial viability kit (Molecular Probes, Eugene, OR, USA).

Bacterial entry and intracellular survival were studied as follows. Trophoblasts were seeded at 3 × 10^5^ cells per well (24-well plates) and myeloid cells at 5 × 10^5^ cells per well and incubated at a multiplicity of infection (MOI) of 50:1. After 4 h, cells were extensively washed to discard unbound bacteria and the number of bacteria associated with cells was determined by quantitative PCR (see below). Bacterial survival was determined at 4 h of incubation. Then infected cells were extensively washed at 4 h post-infection to discard unbound bacteria and incubated for 12 additional days, with bacterial survival evaluated every 3 days.

Cell infection was also illustrated by confocal microscopy. In brief, paraformaldehyde-fixed cells were permeabilized with 0.1% Tween 20 for 10 min, and incubated with rabbit antibodies directed against anti-*C. burnetii* (dilution: 1/500) for 1 h followed by 1:500 anti-IgG rabbit antibodies coupled with Alexa Fluor 555 (ThermoFisher Scientific, Waltham, MA, USA). Cells were also labelled with 1:500 4′,6-diamidino-2 phenylindole (DAPI, Invitrogen) to reveal nuclear material and 1:50 phalloidin (Life Technologies) to reveal polymerized actin for 30 min. Fluorescence signals were analysed by an LSM800 Airyscan confocal microscope (Zeiss) and a 63 × oil objective.

### 4.4. Inflammatory Cytokine Response of Myeloid Cells

To study cytokine transcription myeloid cells were infected with *C. burnetii* organisms (50 bacteria per cell) for 6 h. RNA extraction was carried out with RNeasy MiniKit (Qiagen, Courtaboeuf, France) and real-time quantitative PCR (RT-PCR) was performed using the Moloney murine leukaemia virus reverse transcriptase (MMLV)-RT kit (ThermoFisher Scientific) as previously described [22]. The expression of genes encoding inflammatory cytokines (IL1β, TNF, IL6) and the gene encoding the anti-inflammatory cytokine (IL10) was evaluated by real-time quantitative PCR using specific primers (Table 3) and Smart SYBR Green fast Master kit (Roche Diagnostics, Meylan, France). Ct values were obtained from a CFX Touch Real-Time PCR detection system (Bio-Rad, Nydalsveien, Oslo, Norway) and were normalized using the housekeeping ACTB gene encoding β-actin. The results were expressed as fold change (FC) of investigated genes. FC= 2^−ΔΔCt^ where ΔΔCt = Ct_infected_ − Ct_uninfected_, as previously described [44].

We also used specific immunoassay kits to study the release of inflammatory and immunoregulatory cytokines. For that purpose, macrophages were incubated with *C. burnetii* organisms (50 bacteria per cell) for 24 h. Cells and cell debris were discarded after centrifugation and supernatants were kept at −80 °C before cytokine determination. The sensitivity of the IL-1β kit (R&D Systems, Minneapolis, MN, USA) was 0.125 pg/mL and those of TNF (Life Technologies), IL-6 (Clinisciences, Nanterre, France) and IL-10 (Becton Dickinson Biosciences, Le Pont-de-Claix, France) were 5.5 pg/mL, 15.4 pg/mL and 3.9 pg/mL, respectively.

### 4.5. C. burnetii Infection of Placental Explants

Placental tissue from the basal to the chorionic plates from the region closest to the umbilical cord of four placentas was excised [41]. Three sections (of about 5 mm^2^) were separated in decidua basalis (DB), inner chorionic villi (iCV) and chorionic villi to the foetal side (CVF). These sections were washed and placed in 24-well plates in DMEM F-12 containing 10% FBS, 100 units/mL penicillin, 100 μg/mL streptomycin and 50 mg/mL gentamicin (Life Technologies) for 24 h. The explants were then incubated with 1 × 10^7^ bacteria for 4 h (day 0), washed to eliminate free bacteria and incubated for 8 days in DMEM-F12 containing 10% FBS, 100 units/ml penicillin, 100 μg/ml streptomycin and 50 mg/ml gentamicin. Every two days new medium was added to the explants. Bacteria entry was evaluated at day 0 (4 h) and survival every two days. Three replicates were used per placental section per donor.

### 4.6. In Vivo Experiments

Female BALB/c mice from 5–8 weeks were obtained from Charles River laboratories (Saint-Germain-Nuelles, France). Five mice per group were inoculated with 1 × 10^8^ bacteria or phosphate buffered saline as control via the intraperitoneal route. Body weight and mouse behaviour were evaluated every three days. After 11 days, the mice were euthanized and the blood, heart, lungs, liver, spleen and adipose tissue were collected. Serum was used to determine the antibody response of mice to infection and excised tissues were conserved at −80 °C or fixed in 4% formol for DNA extraction and histological investigations, respectively.

The titer of circulating specific antibodies directed against *C. burnetii* was quantified using immunofluorescence assay, as previously described [44]. In brief, serial dilutions of sera (starting dilutions of 1:25 for IgM and 1:50 for IgG) were added to bacteria deposited on glass slides. Specific antibodies were detected using 1:400 dilution of fluorescein isothiocyanate-conjugated goat anti-mouse IgG (Thermo Fisher) or anti-mouse IgM (Jackson ImmunoResearch Laboratories, West Grove, PA, USA).

Collected tissues were embedded in paraffin and sectioned into 3 µm slices. Tissue sections were stained with hematoxylin-eosin-saffron for lesion identification and the number of granulomas was evaluated using the image analyser SAMBA 2005 (SAMBA Technologies, Alcatel TITN, Grenoble, France) as previously described [25].

### 4.7. Bacteria Quantification

The presence of *C. burnetii* in cells, placental explants and tissues was determined by quantitative PCR. DNA was extracted using the E.Z.N.A Gel Extraction Kit DNA (Norcross, GA, USA) according to the manufacturer’s instructions. Mice tissue was previously digested from 30 mg of starting material before using the DNA tissue extraction kit (OMEGA, Bobigny, France). The infection rate was evaluated using equal DNA amounts from the same tissue or cell type. Quantitative polymerase chain reaction (qPCR) was performed using specific primers (Table 4) as previously described [45]. A control range of bacteria of known concentration was used. The results were expressed as DNA copies for infected cells or tissue biopsies (30 mg of tissue). The samples were considered positive when the qPCR threshold cycles (Ct) were <36.

### 4.8. Genome Sequencing, Comparative Analyses and Phylogeny

Genomic DNA was extracted using the EZ1 biorobot and the EZ1 DNA tissue kit (Qiagen) and then sequenced on a MiSeq sequencer (Illumina, San Diego, France) with the Nextera Mate Pair sample prep kit (Illumina) and Nextera XT Paired End (Illumina), as previously described [46]. The assembly was performed using Spades V. 3.15 [47] and trimmed (using Trimmomatic v. 0.36) [48]. Scaffolds of < 800 bp and scaffolds with a depth value < 25% of the mean depth were removed. The best assembly was selected using criteria such as the number of scaffolds, N50, and number of N. All assembled genomes were annotated using Prokka v 1.14.5 (Fontainebleau, France) [49]. The gff3 output files were used to construct a core genome alignment using Roary v3.13.0 (USA) using default parameters [50]. Comparative analysis was performed for the five placental isolates and reference strains such as RSA493 (Genbank: NC_002971.4), Z3055 9Genbank: NZ_LK937696), NL3262 (Genbank: NZ_CP013667), and Guiana (Genbank: HG825990.30). The phylogenetic tree was reconstructed by using FastTree (Berkeley, CA, USA) [51].

### 4.9. Statistical Analysis

Statistical analysis was performed with GraphPad Prism 6 (Graphpad Software Inc., San Diega, CA, USA). Bacterial entry into cells, transcription and production of cytokines, and the number of bacteria in explants and mice tissue according to bacterial isolates and the reference NM strain were compared using the one-way ANOVA test and Tukey’s multiple comparisons test. Bacteria survival in cells and mouse body weight were analysed using the two-way ANOVA test and Dunnett’s multiple comparisons test. Hierarchical clustering of gene expression was analysed using the ClustVis webtool and an ANOVA test with a Sidak’s multiple comparisons test. Values represent the mean ± standard deviation. The limit of significance was set up at *p* < 0.05.

## Figures and Tables

**Figure 1 ijms-24-01209-f001:**
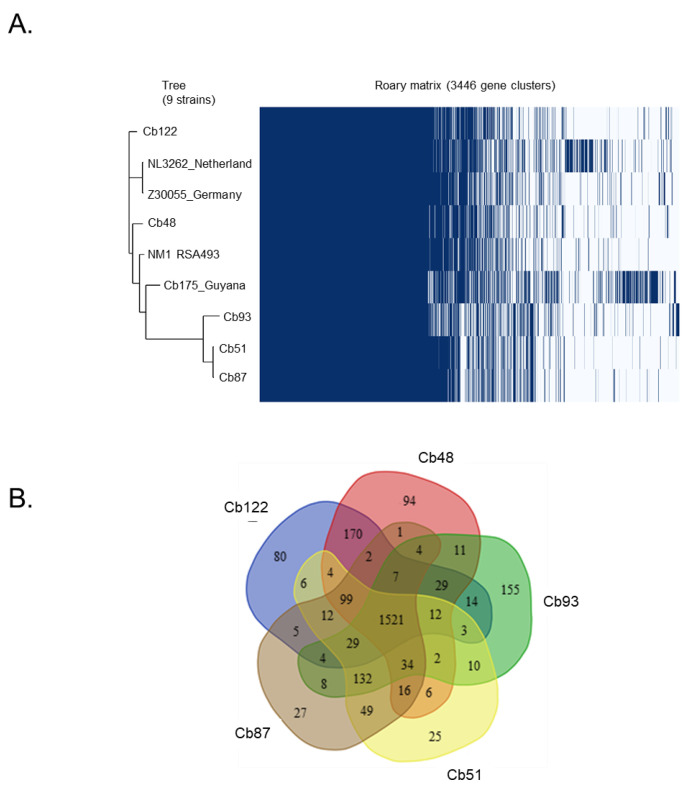
Comparative genomic analyses of placental isolates of *C. burnetii*. (**A**) The genome-based phylogenetic tree was created with the nine investigated isolates; (**B**) a Venn diagram was created to illustrate the core genome and the number of genes specific to *C. burnetii* isolates.

**Figure 2 ijms-24-01209-f002:**
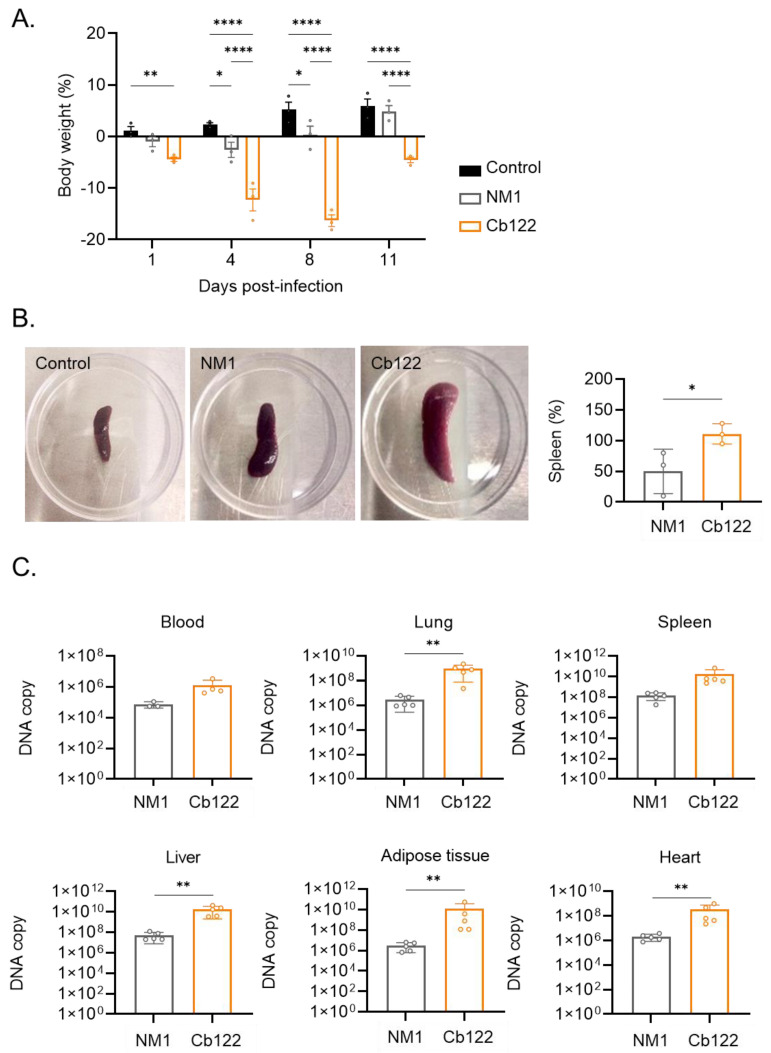
Virulence of *C. burnetii* Cb122 isolate in a murine model of infection. Female Balb/c mice were infected with the *C. burnetii* Cb122 isolate and NM strain (five mice per group). (**A**) The whole body weight of infected mice was monitored every three days. Eleven days post-infection, (**B**) splenomegaly was evaluated as illustrated in the spleen images (left panel), as was the percentage of change relative to the control group (right panel). The graph scale is shown as a logarithm. The bacterial burden was evaluated as (**C**) the number of DNA copies from 30 mg of lung, liver, adipose tissue, spleen, heart and blood. * *p* < 0.05, ** *p* < 0.01, **** *p* < 0.0001.

**Figure 3 ijms-24-01209-f003:**
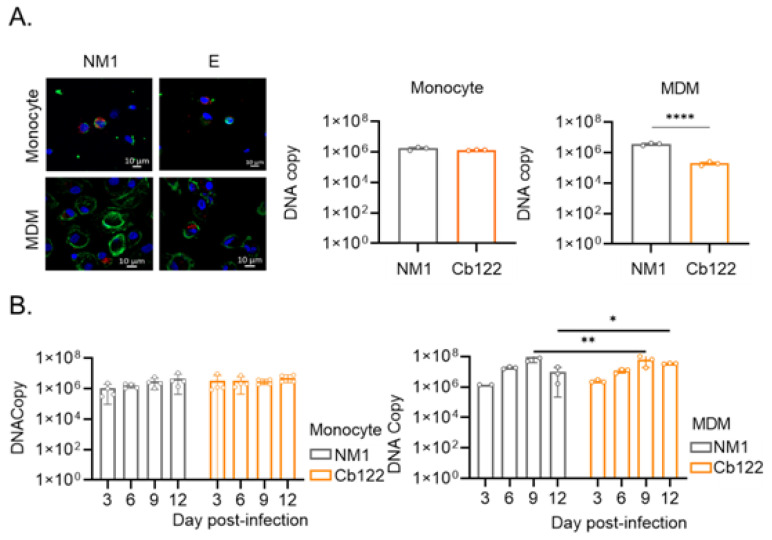
Persistence/replication of *C. burnetii* placental Cb122 isolate in monocytes and MDM. Monocytes and MDMs provided by healthy donors (n = 3) were infected with *C. burnetii* Cb122 isolate (50 MOI). (**A**) Four hours post-infection (p.i.), the bacteria entry was evaluated by immunofluorescence (top panel, F-actin in green, nucleus in blue and *C. burnetii* in red) and quantified by quantitative polymerase chain reaction (bottom panel). (**B**) Survival during the 12-day p.i. period was also investigated through the quantification of DNA copies by qPCR for monocytes (top panel) and MDMs (bottom panel). * *p*< 0.05, ** *p*< 0.01, **** *p* < 0.0001.

**Figure 4 ijms-24-01209-f004:**
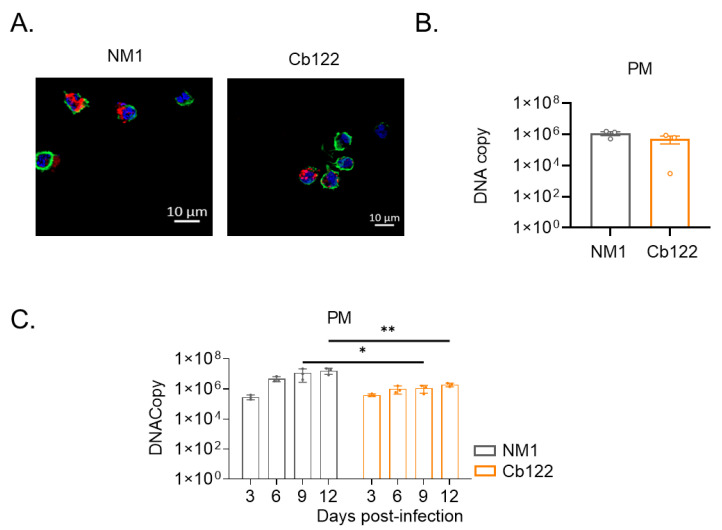
Placental macrophage infection by *C. burnetii* placental isolates. Placental macrophages (PM) isolated from full-term placenta from healthy donors (n = 3) were infected with *C. burnetii* Cb122 isolate (50 MOI). Four hours post-infection (p.i.), bacteria entry was evaluated by (**A**) immunofluorescence (F-actin in green, nucleus in blue and *C. burnetii* in red) and (**B**) quantified by quantitative-polymerase chain reaction. (**C**) Survival during the 12-day p.i. period was also investigated through the quantification of DNA copies by qPCR. * *p* < 0.05, ** *p* < 0.01.

**Figure 5 ijms-24-01209-f005:**
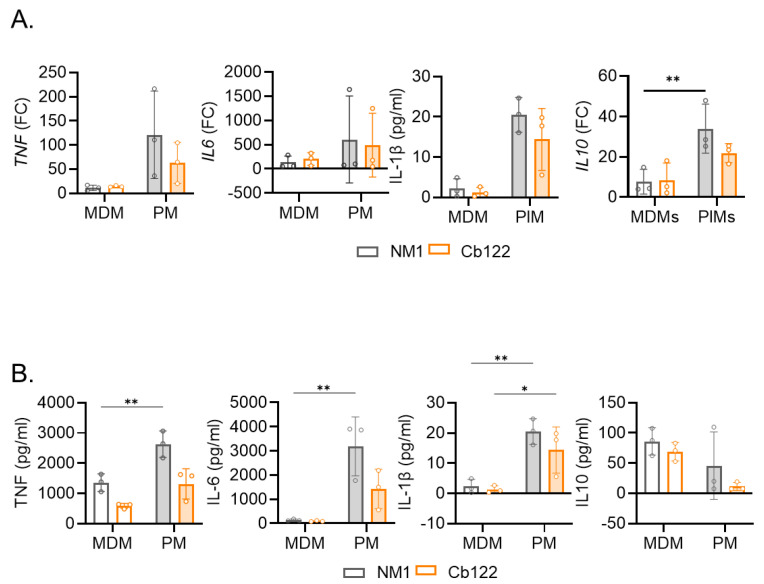
Macrophage inflammatory responses to Cb122 isolate infection. MDMs or placental macrophages (PM) were infected with *C. burnetii* NM (50 MOI) and *C. burnetii* Cb122 isolate (n = 3). After 6 h of infection, the expression of genes involved in the inflammatory (*TNF*, *IL1B*, *IL6*) or immunoregulatory (*IL10*) response was investigated by quantitative reverse-transcription polymerase chain reaction after normalization with the housekeeping actin gene (*ACTB*) as the endogenous control. The data are illustrated as (**A**) the fold change (FC) of the investigated genes. (**B**) After 24 h of infection, the released TNF, IL-1β, IL-6 and IL-10 cytokines were evaluated in the culture supernatants by ELISA assay. * *p* < 0.05, ** *p* < 0.01.

**Figure 6 ijms-24-01209-f006:**
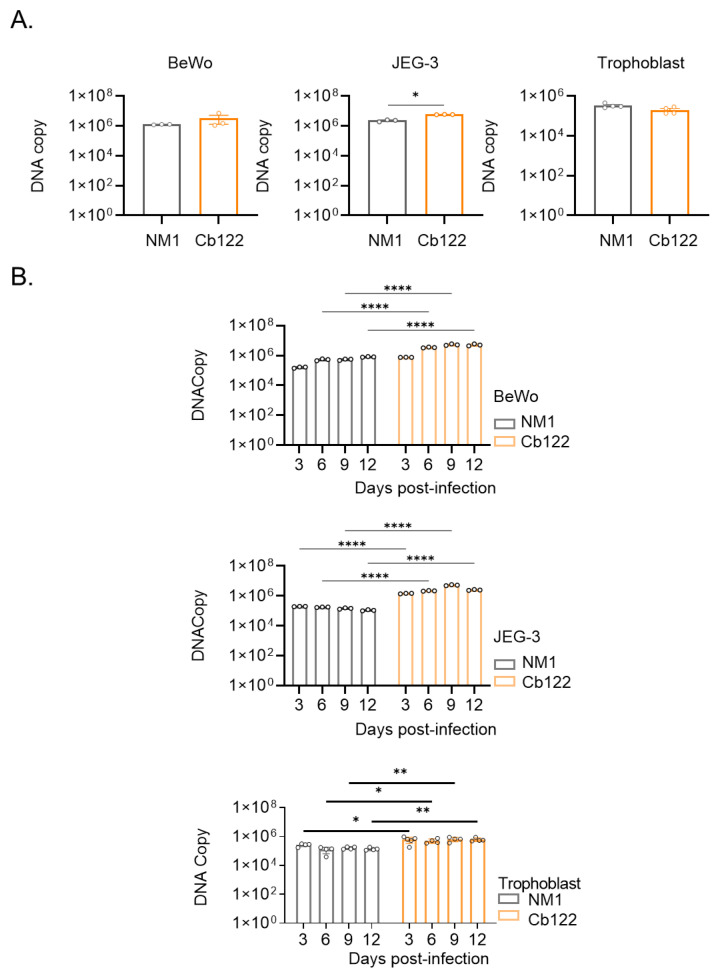
Trophoblast infection by *C. burnetii* placental Cb122 isolate. Human villous trophoblast (BeWo), extra-villous trophoblast (JEG-3) and primary trophoblast provided from healthy donors (n = 3) were infected with the NM strain and *C. burnetii* Cb122 isolate (50 MOI). (**A**) Four hours post-infection (p.i.), bacteria entry was quantified by quantitative polymerase chain reaction. (**B**) Survival during the 12-day p.i. period was also investigated through the quantification of DNA copies by quantitative polymerase chain reaction. * *p* < 0.05, ** *p* < 0.01, **** *p*< 0.0001.

**Figure 7 ijms-24-01209-f007:**
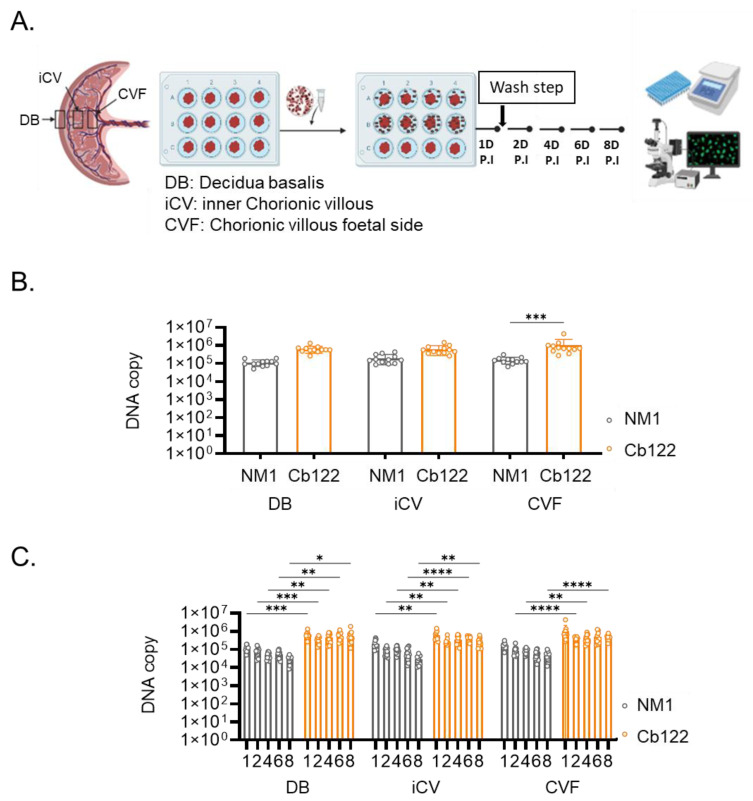
Placental explants and *C. burnetii* isolate infection. (**A**) Placental explants were performed on different sections of placental tissue. Three types of explants were infected with the *C. burnetii* isolate Cb122 and NM strain, and the (**B**) bacteria entry (24 h post-infection) and (**C**) survival up to 8 days post-infection were evaluated by quantitative polymerase chain reaction. * *p* < 0.05, ** *p* < 0.01, *** *p*< 0.001, **** *p*< 0.0001.

**Table 1 ijms-24-01209-t001:** Pan-genomic evaluation (number of genes).

Name	Genome Size (bp)	Elements	Genes (Number)			
			Core	Accessory	Unique	Missed
Cb48	2,000,491	2012	1379	1974	38	1429
Cb51	2,035,629	1960	1379	1939	21	1481
Cb87	2,029,231	1950	1379	1930	20	1491
Cb93	2,048,785	1975	1379	1882	93	1466
Cb122	1,999,784	1997	1379	1953	44	1444
Cb175	1,995,488	2384	1379	2016	368	1057
NL3262	1,995,457	2200	1379	1947	253	1241
RSA493	2,093,477	2003	1379	1962	41	1438
Z3055	1,989,565	2042	1379	1981	61	1399

**Table 2 ijms-24-01209-t002:** *Coxiella burnetii*.

*C. burnetii*	Source (Species)	Isolation (Country/Year)	MST
Cb48	human placenta	France/2012	20
Cb51	human placenta	Spain/2012	4
Cb87	human placenta	France/2012	1
Cb93	human placenta	France/2012	8
Cb122	human placenta	France/2012	22
Nine Mile RSA493 (NMI)	Tick	USA/1935	16

NM: Nine Mile; Cb: *Coxiella burnetii*; MST: multispacer sequence typing.

**Table 3 ijms-24-01209-t003:** Primers used for studying inflammatory gene expression.

Gene	Forward Primer (5′-3′)	Reverse Primer (5′-3′)
*ACTB*	GGAAATCGTGCGTGACATTA	AGGAAGGAAGGCTGGAAGAG
*TNF*	AGGAGAAGAGGCTGAGGAACAAG	GAGGGAGAGAAGCAACTACAGACC
*IL1B*	CAGCACCTCTCAAGCAGAAAAC	GTTGGGCATTGGTGTAGACAAC
*IL10*	GGGGGTTGAGGTATCAGAGGTAA	GCTCCAAGAGAAAGGCATCTACA
*IL6*	CCAGGAGAAGATTCCAAAGATG	GGAAGGTTCAGGTTGTTTTCTG

**Table 4 ijms-24-01209-t004:** Primers used for evaluating *C. burnetii* quantification.

Gene	Forward Primer (5′-3′)	Reverse Primer (5′-3′)
*Com-1*	GCACTATTTTTAGCCG-GAACCTT	TTGAGGAGAAAAACTGGATTGAGA
*C. burnetii* 16S	ACGGGTGAGTAATGCGTAGG	GCTGATCGTCCTCTCA-GACC
Probes	6-FAMGCAAAGCGGGGGATCTTCGG-TAMR	

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
