# Peer review of "Infection and Persistence of *Coxiella burnetii* Clinical Isolate in the Placental Environment"

_ijms, 2023, doi:10.3390/ijms24021209_

Round 1
Reviewer 1 Report
The title is not explanatory about what the work talks about, it would be more correct if it were more specific.
All species of bacteria must be written in italics.
On line 54, further develop the subject of vaccination.
Lines 118-123, we talk about the significance of results without providing the p value.
Regarding the figures:
· The figure 1A doesn't show up well.
· They must be more self-explanatory or develop their content more adequately in the text.
· Is it possible to remove the statistical analysis from the figure captions and make your own section?
@ is used instead of "alpha" on lines 295, 296, and 305.
Author Response
Please find attached the response to the reviewer 1

Reviewer 2 Report
The topic is interesting, but the authors should organize their results as well as the discussion to show how interesting this work is. Authors should edit the English. The authors should emphasize about their findings.
Many results are observed, that is good, but unfortunately there is no sequence when showing these results, when the animal model is used or when they are with cells. This makes it difficult to read the manuscript. It is recommended to organize results, and follow the same sequence in the discussion.
The introduction does not contain the research question, as well as the hypothesis of the work.
In the results of Figure 2, it is not shown how the control is when the DNA copies are indicated.
It is not clear what the authors wanted to show when they talk about DNA copies, what do they mean by this data, what do they want to explain with this?
In Figure 5, they did not evaluate a control?
In the results session, the authors are missing a better description of their findings. They mix up with discussion and results
The authors in the discussion talk in line 267-268: Among genomic specificities of the Cb122 isolate, we reported the presence of an alpha-hemolysin translocation ATP-binding protein (HlyB). Where are these results?
It is not clear, when the authors put forward this hypothesis they speak in the discussion, lines 299-300: However, these results do not support the hypothesis that placenta macrophages are specifically targeted by placenta isolates.
It is recommended to organize the discussion, the authors talk about infections, the presence of LPS, but they do not measure any of this. It is not very clear what they want to highlight.
Author Response
Please find attached the response to the reviewer 2
